# An Eleven-microRNA Signature Related to Tumor-Associated Macrophages Predicts Prognosis of Breast Cancer

**DOI:** 10.3390/ijms23136994

**Published:** 2022-06-23

**Authors:** Sharmilla Devi Jayasingam, Marimuthu Citartan, Anani Aila Mat Zin, Timofey S. Rozhdestvensky, Thean-Hock Tang, Ewe Seng Ch’ng

**Affiliations:** 1Department of Clinical Medicine, Advanced Medical and Dental Institute (AMDI), Universiti Sains Malaysia, Kepala Batas 13200, Penang, Malaysia; sharmilla_487@hotmail.com; 2Department of Biomedical Science, Advanced Medical and Dental Institute (AMDI), Universiti Sains Malaysia, Kepala Batas 13200, Penang, Malaysia; citartan@usm.my (M.C.); tangth@usm.my (T.-H.T.); 3Department of Pathology, School of Medical Sciences, Universiti Sains Malaysia, Kubang Kerian 16150, Kelantan, Malaysia; ailakb@usm.my; 4Medical Faculty, Core Facility Transgenic Animal and Genetic Engineering Models (TRAM), University Muenster, 48149 Muenster, Germany; rozhdest@uni-muenster.de; 5Faculty of Applied Sciences, AIMST University, Bedong 08100, Kedah, Malaysia

**Keywords:** tumor-associated macrophages, M1, M2, miRNA, breast cancer, prognostic biomarker, miRNA-21, miRNA-146a

## Abstract

The dysregulation of microRNAs (miRNAs) has been known to play important roles in tumor development and progression. However, the understanding of the involvement of miRNAs in regulating tumor-associated macrophages (TAMs) and how these TAM-related miRNAs (TRMs) modulate cancer progression is still in its infancy. This study aims to explore the prognostic value of TRMs in breast cancer via the construction of a novel TRM signature. Potential TRMs were identified from the literature, and their prognostic value was evaluated using 1063 cases in The Cancer Genome Atlas Breast Cancer database. The TRM signature was further validated in the external Gene Expression Omnibus GSE22220 dataset. Gene sets enrichment analyses were performed to gain insight into the biological functions of this TRM signature. An eleven-TRM signature consisting of mir-21, mir-24-2, mir-125a, mir-221, mir-22, mir-501, mir-365b, mir-660, mir-146a, let-7b and mir-31 was constructed. This signature significantly differentiated the high-risk group from the low-risk in terms of overall survival (OS)/ distant-relapse free survival (DRFS) (*p* value < 0.001). The prognostic value of the signature was further enhanced by incorporating other independent prognostic factors in a nomogram-based prediction model, yielding the highest AUC of 0.79 (95% CI: 0.72–0.86) at 5-year OS. Enrichment analyses confirmed that the differentially expressed genes were mainly involved in immune-related pathways such as adaptive immune response, humoral immune response and Th1 and Th2 cell differentiation. This eleven-TRM signature has great potential as a prognostic factor for breast cancer patients besides unravelling the dysregulated immune pathways in high-risk breast cancer.

## 1. Introduction

Tumor-associated macrophages (TAMs) are macrophages within the tumor microenvironment that play an important role in cancer initiation and progression. These macrophages could be polarized into two main phenotypes with distinct cytokine and chemokine profiles: at one extreme, the M1 phenotype demonstrates proinflammatory and microbicidal/tumoricidal characteristics while at the other extreme, the M2 phenotype shows anti-inflammatory and tumor-promoting characteristics [1]. In breast cancer, TAMs mainly demonstrate the M2 phenotype [2]. It has been shown in many studies that the regulation of breast cancer by TAMs via microRNAs constitutes one of the crucial mechanisms, although the precise interaction requires further elucidation [3].

MicroRNAs (miRNAs) are a large group of small, endogenous non-coding RNAs of 18–23 nucleotides in length. MiRNAs in general negatively regulate gene expression at the post-transcriptional level by mostly binding at the 3′ untranslated region (UTR) of target mRNAs to suppress their expression, although interaction of miRNAs with the 5′ UTR, protein coding sequence and gene promoters has also been reported [4]. Furthermore, miRNAs can positively regulate mRNA expression under certain circumstances [5].

MiRNAs are associated with immunomodulation in cancer progression and regression. Their expression patterns and implications, however, vary in different types of cancer. MiRNAs can either act as tumor suppressors or tumor promoters [6]. In addition, depending on the tumor context, miRNA which acts as a tumor suppressor in certain cancer types may act as a tumor promoter in others [7].

There are two distinct ways that TAMs are associated with miRNAs. First, miRNAs can be transported from cancer cells to TAMs to regulate TAM polarization. Reciprocally, miRNAs derived from TAMs exert their effects on cancer cells to modulate tumor progression [2]. To date, there are limited studies regarding miRNAs related to TAMs and how these TAM-related miRNAs (TRMs) modulate cancer progression, especially in breast cancer. Furthermore, the relationship between these TRMs with known clinicopathological parameters is yet to be explored. Accumulating studies have demonstrated the diagnostic and prognostic value of the miRNA signature in a variety of cancers, including a few on breast cancer [8,9,10,11,12], but none has focused on TRMs. Elucidating the role of these TRMs in breast cancer can help establish a better understanding of the interplay between TAMs and miRNAs in breast cancer progression.

This study aimed to explore the roles of miRNAs in breast cancer in relation to TAMs with a focus on M1/M2 polarization. First, this study curated the list of TRMs by a knowledge-driven literature search. By modelling the different TRM expression profiles, a novel TRM signature of prognostic value, independent of classic clinicopathological parameters, was constructed. Comprehensive analysis of this TRM signature was conducted via enrichment analyses to deduce the underpinning biological processes of TRMs in regulating breast cancer.

## 2. Results

The workflow of this study is summarized in Figure 1 below.

### 2.1. Construction of the Eleven-TAM-Related miRNA Signature

Forty-two TAM-related miRNAs from various cancer studies were identified from a total of 54 related studies (Table 1). The variations of these miRNAs’ precursors available in the TCGA-BRCA dataset (such as let-7a-1, let-7a-2 and let-7a-3 for let-7a) were also included in our analysis. Two of the TRMs (mir-720 and mir-4291) were excluded from further analysis due to the lack of expression data in the TCGA-BRCA dataset, yielding a total of 52 TRM expression profiles. From the UCSC Xena website, a total of 1063 breast cancer cases were extracted from TCGA-BRCA for primary breast cancer after removing duplicate cases and cases with incomplete overall survival (OS) time. These cases were randomly separated into 80% for the training set (*n* = 856) and 20% for the internal validation set (*n* = 207). Pertinent clinicopathological parameters of the training set, validation set and whole cohort are summarized in Appendix A.

Univariate Cox proportional hazards regression analysis showed that 16 out of the 52 TRMs had *p* value < 0.15 in the training cohort (Figure 2A). LASSO-Cox regression analysis ultimately incorporated 11 from the 16 TRMs in the prognostic model of the training cohort (Figure 2B,C), which contributed to the miRNAs signature. The formula of the novel risk score based on this signature was constructed as below:

**Risk score** = 0.263 × (mir-21 expression value) + 0.251 ×(mir-24-2 expression value) + 0.197 × (mir-125a expression value) + 0.169 × (mir-221 expression value) + 0.118 × (mir-22 expression value) + 0.093×(mir-501 expression value) + 0.053 × (mir-365b expression value) + 0.039 × (mir-660 expression value) − 0.243 ×(mir-146a expression value)-0.189 × (let-7b expression value) − 0.166 × (mir-31 expression value).

### 2.2. The Eleven-TAM-Related miRNA Signature Significantly Differentiate the High-Risk Group from the Low-Risk Group

The risk score for each patient was calculated based on the constructed formula (Figure 3). The reliability of the 11-TRM signature was then tested on both training and internal validation cohorts. Kaplan–Meier survival analysis with a two-sided log-rank test in the training cohort showed that patients in the high-risk group had a significantly shorter OS compared to the patients in the low-risk group (*p* value < 0.001) (Figure 4A).

The prognostic value of the signature was further tested in the internal validation cohort whereby a similar significant difference in OS was observed between the high- and low-risk groups (*p* value < 0.05) (Figure 4B). AUCs of time-dependent ROC curves at 5-year OS were 0.69 (95% CI: 0.61–0.77) and 0.66 (95% CI: 0.52–0.81) for the training and internal validation cohorts, respectively (Figure 4C,D). For the whole cohort, patients of the high-risk group had significantly shorter OS (*p* value < 0.001), and the AUCs of time-dependent ROC curves at 3-, 5- and 10-year OS were 0.68 (95% CI: 0.61–0.76), 0.68 (95% CI: 0.62–0.75) and 0.75 (95% CI: 0.66–0.84), respectively (Appendix A). The C-index for the risk score of the whole cohort was 0.68 (95% CI: 0.63–0.74).

### 2.3. Validation of Prognostic Significance of the Eleven-TAM-Related miRNA Signature in the GEO Dataset

The prognostic value of this TRM signature was further validated in the GEO dataset, GSE22220 which contains a cohort of 207 primary breast cancer cases. A similar significant difference in distant-relapse-free survival (DRFS) was observed between the high- and low-risk groups (*p* value < 0.001) (Figure 5A). The AUCs of time-dependent ROC curves at 3-, 5- and 8-year DRFS were 0.54 (95% CI: 0.42–0.66), 0.60 (95% CI: 0.51–0.69) and 0.63 (95% CI: 0.55–0.71), respectively (Figure 5B). The C-index for the risk score for this cohort was 0.58 (95% CI: 0.53–0.65).

### 2.4. Prognostic Significance of the Eleven-TAM-Related miRNA Signature Is Independent of Clinicopathological Parameters

The association analysis between the risk score and pertinent clinicopathological parameters in the TCGA-BRCA dataset is summarized in Appendix A. The risk score was significantly associated with age, gender, pathological stage, histological type, ER and HER2 status and PAM50 (all *p* values < 0.05).

The prognostic value of this signature was further scrutinized among breast cancer subtypes based on ER and HER2 status. The high-risk group had significantly poorer prognosis as compared to the low-risk group in the ER+HER2−and ER−HER2− cohorts (*p* value < 0.01) but not in the ER+/− HER2+ cohort (Appendix A). Interestingly, the risk scores based on the TRM signature was inversely correlated to the scores of TIL in the TNBC cases (Spearman rho = −0.33, *p* value = 0.001, Appendix A).

Univariate Cox proportional hazards regression analysis showed that among the pertinent clinicopathological parameters, higher pathological stages (stage III and IV), ER and HER2 negative status, higher age and high-risk group based on the eleven-TRM signature were poor prognostic factors (*p* value < 0.001, *p* value = 0.004, *p* value < 0.001 and *p* value < 0.001, respectively) (Figure 6A). Multivariate Cox proportional hazards regression analysis revealed that the risk score based on the eleven-TRM signature remained as an indicator for poor prognosis (*p* value < 0.001) with a hazard ratio of 2.28 (95% CI: 1.45–3.58) (Figure 6B).

A nomogram was constructed integrating all the independent prognostic factors in the multivariate analysis. The AUC of the time-dependent ROC curve at 5-year OS for the nomogram-based prediction model was the highest at 0.79 (95% CI: 0.72–0.86). The AUC of time-dependent ROC for the risk group (0.62 (95% CI: 0.55–0.69)) was comparable to the AUCs of other independent prognostic factors (Age: 0.66 (95% CI: 0.58–0.74); pathological stage: 0.67 (95% CI: 0.6–0.75); and ER and HER2 status: 0.61 (95% CI: 0.53–0.68)) (Appendix A). The C-index for the nomogram was the highest at 0.82 (95% CI: 0.77–0.86). C-indices for other factors were risk group: 0.78 (95% CI: 0.69–0.87); Age: 0.66 (95% CI: 0.60–0.73); Pathological stage: 0.80 (95% CI: 0.73–0.88); and ER and HER2 status: 0.65 (95% CI: 0.56–0.74).

### 2.5. Among the Different Immune Infiltrate Populations, TAMs Had the Highest Correlation with the Risk Score Responsible for Poor Prognosis

The risk score based on the TRM signature is validated through high correlations with the abundance of TAM subtypes in the dataset. Among the populations of immune infiltrates estimated from the CIBERSORT algorithm, M2 macrophages showed the highest positive correlation with the risk score (r = 0.241, *p* value < 0.001), whereas CD8 T cells showed the highest negative correlation with the risk score (r = −0.263, *p* value < 0.001), followed by CD4 T cells and M1 macrophages (Figure 7). However, inter-relationship between individual miRNAs and different immune infiltrate populations showed various strengths and directions of correlations without a distinctive pattern with certain immune cell types (Appendix A).

### 2.6. Analysis of the Differential Gene Expression and Gene Set Enrichment

As depicted in Figure 8 below, differential gene expression revealed 4 upregulated genes and 59 downregulated genes in the high-risk group as compared to the low-risk group (adjusted *p* value <  0.05 and |Log2(fold change)| > 1). To gain insights into the underlying biological processes, overrepresentation analysis of the differentially expressed genes was performed using Gene Ontology (GO) to determine the association of the gene sets with any biological processes. No biological process was found to be associated with the upregulated genes. On the other hand, downregulated genes were strongly associated with immune responses such as classical pathway of complement activation, adaptive immune response and humoral immune response (Figure 9).

Genes from the differential expression analysis of coding mRNA data were ranked as described in the method for gene set enrichment analysis. The KEGG pathway gene set enrichment analysis showed that immunity pathways such as Th1 and Th2 cell differentiation, JAK-STAT signaling pathway and Th17 cell differentiation were suppressed, whereas pathways such as proteosome, DNA replication, oxidative phosphorylation and base excision repair were activated in the high-risk group (Figure 10). Gene set enrichment analysis using Hallmark gene sets displayed 20 significantly enriched gene sets; activated gene sets in the high-risk group included Myc Target V2, oxidative phosphorylation and E2F Targets whereas suppressed gene sets included UV response and KRAS signaling (Appendix A).

### 2.7. Relationship between miRNAs in TRM Signature and Differentially Expressed Genes between High- and Low-Risk Groups

A total of 387 and 61 regulatory miRNAs were obtained from TarBase 8.0 based on the downregulated and upregulated genes in the high-risk group. Based on the premise that miRNAs act as negative regulators of target mRNAs, an intersection was performed with the miRNAs in the TRM signature with positive and negative coefficients, respectively. Among these regulatory miRNAs, six and two mature miRNAs intersected with the miRNAs in the TRM signature with positive and negative coefficients, respectively (Figure 11A,C). Similarly, a total of 6736 and 8173 genes were obtained as predicted target mRNA based on miRNAs in the TRM signature with positive and negative coefficients, respectively. Six and one mRNAs intersected with the downregulated and upregulated genes in the high-risk group, respectively (Figure 11B,D). Specifically, six miRNA: RNA pairs were identified based on the miRNAs from TRM signature with positive coefficients and downregulated genes, i.e., hsa-mir-21-3p: IL33, hsa-mir-21-3p: SELP, hsa-mir-22-5p: CLEC10A, hsa-mir-21-5p: TP63, hsa-mir-365b-3p: PTPRT and hsa-mir-22-5p: GP2. Likewise, two miRNA: RNA pairs were identified from the miRNAs in TRM signature with negative coefficients and the upregulated genes. They were hsa-let-7b-5p: PSCA and hsa-mir-146a-5p: PSCA.

## 3. Discussion

Tumor-associated macrophages (TAMs) as immune cells residing within the tumor microenvironment have garnered much interest due to their roles in modulating the progression of breast cancer. Of particular interest is the polarization status of the TAMs that would result in either the suppression or promotion of breast cancer progression. Although numerous studies have proved the vital role miRNAs plays in breast cancer progression, studies that describe the interaction between TAM-related miRNAs (TRMs) and breast cancer progression are scarce. Driven by this scarcity, this study aimed to explore the relationship between TRMs and breast cancer prognosis.

A total of 42 TRMs were identified from the literature search; 9 were associated with breast cancer only, 27 were reported in other cancer types while 6 were associated with both breast and other cancer types. Intriguingly, several TRMs have been reported in more than one cancer type, either exerting similar or total opposite effects. We hypothesized that some TRMs reported in other cancer types may also contribute to breast cancer prognosis, and thus, these TRMs were included although the focus of this study is on breast cancer. Including the precursors of these miRNAs’ genes available in the TCGA-BRCA dataset, a total of 52 TRM expression were analyzed. From the analysis of 52 TRM expression, 11 of them eventually constitute the TRM signature, which was shown to have a significant independent prognostic value for breast cancer (*p* value < 0.005). Among these 11 TRMs, only 4 have been reported in TAMs of breast cancer, namely let-7b, miR-21, miR-24-2 and miR-146a (Table 1). Although there were reports of miR-22, miR-31, miR-125a, miR-221, miR-501 and miR-660 involved in breast cancer [67,68,69,70,71,72], they have not been associated with TAMs so far. MiR-365b, on the other hand, has not been reported in any breast cancer studies to date, but is known to be secreted by M2 TAMs to promote hepatocellular carcinoma cell migration and invasion [73].

Eight of these TRMs had positive coefficients in the constructed TRM signature, which implies worse prognosis based on the expressions of these TRMs. They were miR- 21, miR-24-2, miR-125a, miR-221, miR-22, miR-501, miR-365b and miR-660, arranged in the descending order of their coefficients. Estimated at 0.263, miR-21 had the highest positive coefficient and the highest overall coefficient in the TRM signature, suggesting that miR-21 expression exerts the most impact on breast cancer progression in this cohort. In fact, miR-21 is well-established as the main tumor promoter in various cancers, and multiples studies have evidenced its linkage to poor prognosis. Li et al., in 2017 demonstrated that in breast cancer, the upregulation of miR-21 in TAMs favored M2 transformation and promoted metastasis by inhibiting the expression of *PTEN*, a tumor suppressor gene [25]. Another study by Zheng et al., 2017 revealed that miR-21 derived from M2 TAMs promoted cisplatin resistance in gastric cancer cells. They further demonstrated that miR-21 were transferred from TAMs to gastric cancer cells to suppress cell apoptosis and enhance the activation of the PI3K/AKT signaling pathway via the downregulation of PTEN [23]. Acting as the crucial factor that promotes the tumorigenesis of breast cancer, miR-21 can be potentially targeted to debilitate tumor growth.

The miR-24-2 belongs to the miR-23a/27a/24-2 cluster associated with multiple diseases [74,75]. In a breast cancer study, the miR-23a/27a/24-2 cluster was suggested to work as a double feedback loop. This cluster was downregulated by M1-type stimulation and upregulated by M2-type stimulation. The down- regulation of the cluster is mediated by the binding of the transcription factors involved in macrophage polarization, NF-κB and STAT-X, to the promoter of the miR-23a/27a/24-2 cluster, thus repressing its expression. On the other hand, the upregulation of the cluster was promoted by the binding of STAT6 to the promoter of the cluster. However, this cluster was surprisingly downregulated in TAMs of breast cancer patients, and further probing is needed to determine the cause [27]. Consistent with these findings, miR-24-2 in our study contributes to the worse prognosis of breast cancer. Furthermore, miR-24-2 was highly expressed in the high-risk group of breast cancer patients.

Likewise, miR-221 derived from M2 TAMs promoted cancer cell proliferation in epithelial ovarian cancer (EOC) via suppression of the cyclin-dependent kinase inhibitor 1B (CDKN1B) [49] and aggravates the growth and metastasis of cancer cells in osteosarcoma by targeting SOCS3, which then activates the JAK2/STAT3 pathway [50]. TAM-derived exosomal miR-501 promotes progression of pancreatic ductal adenocarcinoma by inhibiting the tumor suppressor, TGF-beta Receptor III (*TGFBR3*) gene and activating the TGF-β signaling pathway [60]. Similarly, miR-660 was found to be upregulated in exosomes secreted by TAMs of epithelial ovarian cancer [63].

There were limited data on TAM-related miR-125a and miR-22 in cancers, but the transfer of miR-125a or miR-125b to TAMs can suppress cancer cell proliferation and stem cell properties by targeting CD90, a stem cell marker for hepatocellular carcinoma [36]. Meanwhile, miR-22 transferred from TAMs to glioma stem cells was able to promote mesenchymal phenotypes and induce radiotherapy resistance by targeting the chromodomain helicase DNA-binding protein 7 (CHD7), a chromodomain enzyme that maintains the proneural phenotype in glioblastoma [26].

In descending order, miR-146a, let-7b and miR-31 had negative coefficients in this signature, implying their role as tumor suppressor. The miR-146a has many conflicting findings in regard to its functions. Some research groups found miR-146a to act as a tumor suppressor while others stress the tumor-promoting property of this miRNA in breast cancer [7]. In 2015, Li et al., discovered that miR-146a expression was significantly downregulated in the TAMs of patients’ breast tumors. They also reported that inhibiting miR-146a promoted the M1 TAM expression but decreased M2 TAM expression and suppressed tumor growth in mice, indicating the oncogenic function of this miRNA. However, the in vivo studies on mice revealed that the decreased expression of miR-146a in macrophages somehow also inhibited tumor growth at the same time. The authors acknowledged that miR-146a function in TAMs appeared to be contradictory to the observation that miR-146a was downregulated in TAMs of breast cancer and postulated that miR-146a is a negative regulator in TAM polarization [41]. In this study, our TRM signature showed that miR-146a acts as a tumor suppressor, associating it with good prognosis. In fact, miR-146a possessed the highest negative coefficient with a value of −0.243 and hence constitutes the most potent tumor-suppressing function of our signature. This was substantiated by our finding that miR-146a was highly expressed in the low-risk group compared to the high-risk in our cohort. The contradictory role portrayed by miR-146a in our study compared to the ones reported in other studies seems to suggest the involvement of other alternative pathways or mechanisms that may influence its function and needs further investigation.

Let -7b is a known tumor suppressor in breast, gastric and ovarian cancers [76,77,78]. In 2016, Zhen Huang and his team discovered that the administration of let-7b in tumor cells can repolarize M2 TAMs to M1, reverse the suppressive tumor microenvironment and inhibit tumor growth in a breast cancer mouse model [14], consistent with the tumor suppressive signature displayed by let-7b in our TRM panel. To the best of our knowledge, there were no studies on TAM-related miR-31 in breast cancer, although miR-31 expression was significantly downregulated in many cancers, including breast, ovarian, prostate [79] and gastric cancers [80].

In this study, we have successfully constructed an eleven TAM-related miRNA-based signature that was significantly associated with OS/ DRFS in breast cancer patients. The performance of this signature was validated in both TCGA and GEO cohorts whereby Kaplan–Meier analyses demonstrated a significant difference in survival between the low- and high-risk groups. The performance of this signature is on par with other well-established multi-gene based prognostic tools such as Endopredict and Oncotype Dx, which had C-indices ranging from 0.6 to 0.7 [81]. In comparison, C-indices of the risk score based on the TRM signature in our TCGA-BRCA and GSE22220 cohorts were 0.68 and 0.58, respectively. As C-index is not a proper metric to predict a t-year risk of an event [82], AUC metrics for time-dependent ROC curves at a specific t-year OS/DRFS were chosen in our study. The prognostic value of this signature was further enhanced with the amalgamation of independent clinical factors, resulting in the highest AUC of the nomogram for the 5-year OS at 0.79. In addition, when considering all the independent prognostic factors, the risk group based on this TRM signature has similar prognostic performance with other traditional established prognostic factors either in terms of AUC metrics or C-indices. This signature in combination with other relevant independent prognostic factors has a promising potential in the prognostication of breast cancer.

The fact that the tumor microenvironment (TME) components and functions differ across the various subtypes of breast cancer is well-established [83]. Since the abundance of TAMs varies in different subtypes of breast cancer, the risk scores of these cases also varied accordingly (Appendix A). The prognostic value of risk score based on the TRM was demonstrated in the whole TCGA-BRCA cohort, and further sub-group analysis showed that its prognostic value was retained especially in the ER+HER2− and ER−HER2− cohorts. Intriguingly, risk score derived from this signature was inversely corelated with the TIL score in TNBC cases. It was known that high TIL scores in TNBC cases are associated with better response to chemotherapy and improved overall survival [88]. Therefore, the inverse relationship of this risk score with TIL score could be alluded to in part the underlying interaction between TAMs and TILs in TNBC cases.

Analysis of the immune infiltrate estimates from the CIBERSORT algorithm revealed that among the different immune infiltrate populations, the constellation of eleven miRNAs as a risk score had a collective relationship with TAMs. However, analysis between individual miRNAs in the TRM signature and different immune infiltrate populations did not recapitulate such relationship at the individual miRNA level. From the chord diagram in Appendix A, several miRNAs did show significant correlation with M1 or M2 TAMs; nonetheless, they were also correlated with other immune cell types concurrently, forming many-to-many relationships between miRNAs and different immune infiltrate populations. The relationship of these miRNAs with TAMs is thus best alluded to the collective effects by these miRNAs, presumably via a complex regulatory network rather than simple additive effects by individual miRNAs. One such outcome of the complex regulatory network could be the Th1-Th2 cell differentiation, which was shown to be significantly suppressed in the high-risk group by KEGG pathway gene set enrichment analysis. Taken together, such a result reflects the complex intertwined relationships among miRNAs, TAM polarization and the Th1/Th2 paradigm.

Bioinformatics analysis was applied to elucidate the biological functions of this TRM signature. From the differential gene expression analysis, four genes were upregulated while 59 genes were downregulated in the high-risk group. GO analysis showed that the downregulated genes were strongly involved in immune pathways, such as adaptive immune response, humoral immune response and immune response signaling pathway (*p* value < 0.001). It is well-established that immune pathways detect and destruct cancerous cells. Adaptive immune response, for one, consists of T cells, B cells and antigen-presenting cells that target and kill antigens specific to the cancer cells [84]. Similarly, humoral immunity develops autoantibodies against tumor-associated proteins. In 2020, Sato et al. discovered that humoral immunity plays a vital part in the suppression of breast cancer. In a cohort of 500 invasive breast cancer patients, they found that the recurrence-free survival of the high anti-HER2 autoantibody (HER2-AAb) group was significantly longer than that of the low HER2-AAb group (*p* value = 0.015). The high HER2-AAb group also had a higher number of CD20, IGKC and CXCL13 immune cells, indicating enhanced humoral immunity compared to the low HER2-AAb group [85]. In short, GO analysis suggests that the eleven miRNAs were indeed key players in modulating breast cancer via immune pathways. Therefore, immunomodulatory strategy in concert with the targeted inhibition of miRNAs can be employed to ameliorate tumorigenesis and cancer progression.

Furthermore, KEGG analysis revealed that mechanisms such as DNA replication, base excision repair, proteosome and oxidative phosphorylation were significantly activated in the high-risk group while immunity pathways such as Th1 and Th2 cell differentiation, JAK-STAT signaling pathway and Th17 cell differentiation were suppressed. Proteosome complexes support cancer cell differentiation and survival [86], while oxidative phosphorylation is rudimentary in cancer cell proliferation, stemness and metastasis [87]. In fact, a recent paper revealed that improved anti-tumor response was observed in TNBC cell lines and in patient-derived tumor xenograft models when marizomib, a proteasome inhibitor derived from marine bacteria was used to inhibit the multiple proteasome catalytic activities and oxidative phosphorylation in vivo [88].

Th1, Th2 and Th17 cells are all subsets of the CD4+ T cell. Th1 cells produce tumor necrosis factor alpha (TNF-*α*), interferon gamma (IFN-γ), interleukin (IL)-2 and IL-12 to mediate antitumor effects, while Th2 cells produce IL-4 and IL-10 which favor tumor growth [89]. Hence, repressing the Th1/Th2 balance would promote tumor growth. Eftekhari et al., in 2017, discovered that Th17 cell markers were significantly decreased in stage IV of breast cancer [90]. Our findings verified that a decrease in Th17 correlates with worse prognosis.

The relationship between miRNAs in TRM signature and differentially expressed genes between high- and low-risk groups was explored via TarBase 8.0, a reference database cataloguing experimentally supported miRNA targets [91]. This exploration revealed six miRNA: mRNA pairs when a higher expression of the miRNAs with positive coefficients intersected with downregulated genes in the high-risk group. Intriguingly, among these downregulated genes, CLEC10A and PTPRT have been demonstrated as poor prognostic factors when their expressions were reduced in breast cancer, suggesting a plausible negative regulatory mechanism of these miRNAs in our TRM signature [92,93]. Similarly, it reduced the expression of the TP63; a known suppressor of cell migration and metastasis and could lead to enhanced tumor invasion and migration [94].

When miRNAs with negative coefficients intersected with upregulated genes, two miRNA: mRNA pairs targeted the same PSCA gene, i.e., hsa-let-7b-5p: PSCA and hsa-mir-146a-5p: PSCA. Our results show that reduced let-7b and mir-146a expression could account for a higher expression PSCA, which has been shown to correlate with unfavorable histological features and HER2/neu overexpression in breast cancer, although there was no association of PSCA with patients’ prognosis [95].

Another mechanism of TAM-mediated cancer progression worth noting is via the methylation of breast cancer-specific genes to regulate gene expression [96]. A recent paper discovered that TAMs secrete IL-6 in the TME to stimulate protein arginine methyltransferase 1 (PRMT1)-mediated meR342-EZH2 formation in order to stabilize the enhancer of Zeste Homolog 2 (EZH2) in breast cancer cells. The EZH2 enrichment subsequently enhanced breast cancer metastasis. The detailed mechanism behind the increment of PRMT1, however, remains unknown [97]. It would be interesting to see in the future if our signature plays any part in this TAM-mediated methylation.

We acknowledge the existence of other miRNA-based prognostic models in breast cancer [98,99], including the seminal work by Iorio et al., which first showed differentially expressed miRNAs in breast cancer correlated with specific clinicopathological features [100]. However, our approach is unique, where instead of comparing the differentially expressed miRNAs in cancer and normal breast tissue, we searched for miRNAs from published literatures that have been proven by experimental data. Furthermore, this study focused on TAM-related miRNAs instead of miRNAs in general, hence narrowing the target for future anti-cancer therapies. In addition, gene set analyses in most studies were based on the list of bioinformatics-predicted mRNAs from the corresponding relevant miRNAs [98,99,101], whereas in this study, we performed the differential expression of coding mRNAs based on risk stratification by the TRM signature to identify the truly differentially expressed genes attributed to this signature.

Overall, as a retrospective study, our data may have certain limitations and would certainly benefit from prospective experimental validation.

## 4. Methods

### 4.1. Literature Search for TAM-Related miRNAs

Literature search was performed with the keywords: “miRNA”, “tumor-associated macrophage”, “macrophage” and “cancer” from the year 2016 to 30 May 2021. The NCBI PubMed database was utilized to identify potential prognostic miRNAs that were either delivered to TAMs or derived from TAMs in various cancer types. The scope was broadened to include TRMs within cancer in general, as there were very few TRMs reported in breast cancer alone. Publications that do not mention TAMs were excluded.

### 4.2. Data Mining of miRNAs, mRNAs and Clinicopathological Data

Stem-loop miRNAs expression and the expression of mRNAs were downloaded from The Cancer Genome Atlas Breast Cancer (TCGA-BRCA) dataset using UCSC Xena [102]. Stem-loop miRNAs expression were miRNA sequencing (miRNAseq) data quantified by a modified version of the profiling pipeline developed by Chu et al. [103]. MiRNAs were annotated based on the miRBase (version 21.0) database. Curated survival data and phenotypes were also obtained. Specifically, data regarding overall survival (OS) and overall survival time (OST), age, gender, estrogen receptor (ER) status, Human Epidermal Growth Factor Receptor-2 (HER2) status, pathological stage, intrinsic subtype based on PAM50 and histological subtype were extracted. The dataset was randomly split into training cohort (80%) and internal validation cohort (20%).

### 4.3. Prognostic TAM-Related miRNA Signature Construction

Analysis was performed using R software version 4.1.0. TRMs were first screened for their prognostic values in the training cohort by univariate Cox proportional hazards regression analysis using the “survival” package. The *p* value was adjusted to <0.15 as these miRNAs were curated from literature supported by experimental data. Then, least absolute shrinkage and selection operator (LASSO)-Cox regression analysis was applied using the “glmnet” package to construct the TRM signature with 5-fold cross validation. The risk score for each patient was computed based on this signature by calculating the sum product of miRNA expressions and their respective coefficients.

Patients were ranked by their risk scores and subsequently assigned into low- and high-risk groups. The median risk score from the training cohort was used as the cut-off point. Kaplan–Meier analysis with two-sided log-rank test using “survminer” package was conducted to determine the prognostic value of the risk score in training, internal validation and whole cohorts with the level of significance set at *p* value < 0.05. To evaluate the prognostic capacity of the risk score based on the TRM signature for overall survival, the time-dependent receiver operating characteristic (ROC) curve was drawn using the “timeROC” package. Subsequently, the Area Under Curve (AUC) metric was calculated for training, internal validation and whole cohorts. The prognostic performance of the risk score for the whole cohort was further evaluated by Harrell’s concordance index (C-index) using the “survcomp” package.

### 4.4. External Validation of the Prognostic Significance of TAM-Related miRNA Signature with the Gene Expression Omnibus (GEO) Dataset

The GEO database was searched for breast cancer datasets using keywords “miRNA” and “breast cancer”, and the search was limited to studies using human samples, non-coding RNA profiling by array/high throughput sequencing, and with a sample count of more than 200. One dataset, GSE22220, was identified as containing 207 primary breast cancer cases with miRNA expression profiled using Illumina Human v1 MicroRNA expression beadchip containing 735 microRNAs designed against miRBase (version 9.1) and potential miRNAs identified in a RAKE analysis study. The miRNA expressions were normalized log2 signal intensities. Available distant-relapse-free survival data with a complete 10-year follow-up was also downloaded. Kaplan–Meier analysis, time-dependent receiver operating characteristic (ROC) curve, AUC metric analysis and C-index were performed to validate the prognostic capacity of the risk score based on the TRM signature in this external dataset. Similarly, the median risk score was used as the cut-off point to assign patients into low- and high-risk groups.

### 4.5. Independent Prognostic Significance of the Risk Score Based on TAM-Related miRNAs Signature

Using the TCGA-BRCA dataset, association analyses between the risk score and other pertinent clinicopathological parameters were conducted using Pearson correlation, Welch *t*-test or one-way ANOVA. Univariate and multivariate Cox proportional hazards regression analyses were conducted using the “survival” package to evaluate the prognostic capacity of the risk score in distinguishing between high-risk and low-risk patients and for its independence from other pertinent clinicopathological parameters. These parameters were age, gender, histological subtype, ER and HER2 status, intrinsic subtype based on PAM50 and pathological stage. The level of significance was set at *p* value < 0.05. A nomogram was constructed based on the multivariate analysis using “rms” package to predict the 3-, 5- and 10- year survival probability. A time-dependent ROC curve was further drawn to evaluate the prognostic capacity of the nomogram-based prediction model for overall survival by the AUC metric for the whole cohort. C-index was also calculated.

The prognostic capability of this signature was further assessed based on ER+HER2−, ER+/− HER2+ and ER−HER2− breast cancer cohorts via Kaplan–Meier analysis. Focusing on triple-negative breast cancer (TNBC) cases, scores of tumor-infiltrating lymphocytes (TIL) of these cases were extracted from a recent study [104]. The relationship between risk score based on the TRM signature and TIL scores was sought using Spearman correlation.

### 4.6. Correlation between Prognostic Risk Score and Immune Infiltrate Populations

Immune infiltrates in the tumor tissue were inferred using a gene expression deconvolution algorithm, CIBERSORT. The data for the TCGA-BRCA dataset were downloaded from TIMER2.0 (http://timer.cistrome.org/ (accessed on 27 April 2022)). The Pearson correlation was performed between the risk score and the population of immune infiltrate estimates by using the “ggcorrplot” package. A Pearson correlation coefficient with a *p* value < 0.05 was considered statistically significant. A Pearson correlation between expressions of individual miRNAs and the populations of immune infiltrate estimates was also performed. The inter-relationship between miRNAs and immune infiltrate populations as visualized by chord diagrams using the “circlize” package. Correlations with correlation coefficients carrying *p* values < 0.01 were used to construct the chord diagrams.

### 4.7. Differential Gene Expression and Enrichment Analysis

The raw count data of mRNA expressions from the TCGA-BRCA dataset were first subjected to the filtration of the low expression genes prior to normalization by the weighted trimmed mean of M-values method in the “edgeR” package, and they were transformed by voom in “limma” package for differential gene expression between the high- and low-risk groups. The differentially expressed genes were defined according to adjusted *p* value <  0.05 and |Log2(fold change)| > 1.

Three types of enrichment analyses were utilized to deduce the biological functions of the differentially expressed genes, which were coding mRNA data from the TCGA-BRCA dataset. First, over-representation analysis of Gene Ontology (GO) to associate the differentially expressed genes with biological processes was performed using the “clusterProfiler” package. Genes were ranked based on the association between their expression and the class distinction (high- or low-risk groups) by a ranking metric, which was defined as the sign of the fold change multiplied by the inverse of the adjusted *p* value obtained from differential expression analysis. Next, the Kyoto Encyclopedia of Genes and Genomes (KEGG) pathway gene set enrichment analysis and gene set enrichment analysis using Hallmark gene sets from The Molecular Signatures Database (MSigDB) v7.4. were performed for the ranked genes. Adjusted *p* value < 0.05 was considered as statistically significant.

### 4.8. Relationship between miRNAs in the TRM Signature and the Differentially Expressed Genes of High- and Low-Risk Groups

To explore the relationship between the miRNAs in the TRM signature and the differentially expressed genes, target mRNAs of the miRNAs in the TRM signature and regulatory miRNAs of the differentially expressed genes in *homo sapiens* were downloaded from miRNet (https://www.mirnet.ca/ (accessed on 27 April 2022)) based on TarBase 8.0 [104]. Venn diagrams were drawn using the “ggVennDiagram” package to display the relationship between the miRNAs in the TRM signature and regulatory miRNAs inferred from the differentially expressed genes. Similarly, target mRNAs of the miRNAs in the TRM signature were compared to the differentially expressed genes between the high- and low-risk groups.

## 5. Conclusions

We have successfully constructed an eleven-TAM-related miRNA-based signature in this study that could act as an independent prognostic factor. With further exploration, this signature has the potential to provide best survival estimates, ease prognostication and guide treatment that targets TAMs and immune-related pathways.

## Figures and Tables

**Figure 1 ijms-23-06994-f001:**
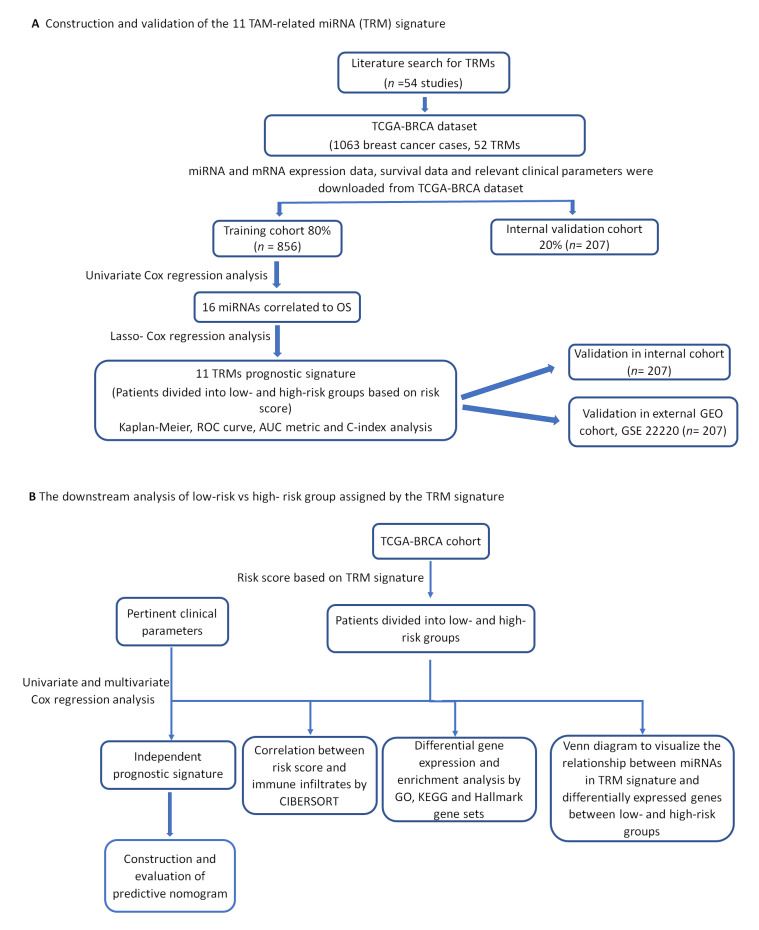
The overall workflow describing the process involved in the construction of 11TAM-related miRNA signature. (**A**) Flow chart describing the process involved in developing and validating the prognostic significance of the 11TAM-related miRNA signature. (**B**) Flow chart showing the prognostic independence evaluation and downstream analysis of high-risk vs. low-risk group assigned by the TRM signature.

**Figure 2 ijms-23-06994-f002:**
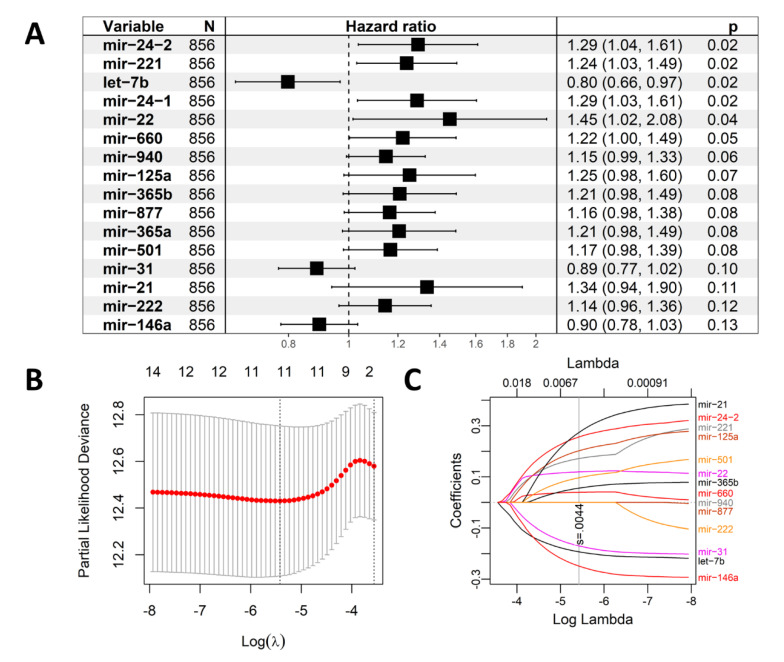
Cox regression analysis. (**A**) The 16 TAM-related miRNAs with *p* value < 0.15 and their hazard ratios from univariate Cox proportional hazards regression analysis. (**B**) Tuning parameter (λ) selection in the LASSO model for OS-relevant miRNAs. (**C**) The LASSO coefficient profile of the 16 miRNAs. The vertical line indicates the coefficient selected by LASSO.

**Figure 3 ijms-23-06994-f003:**
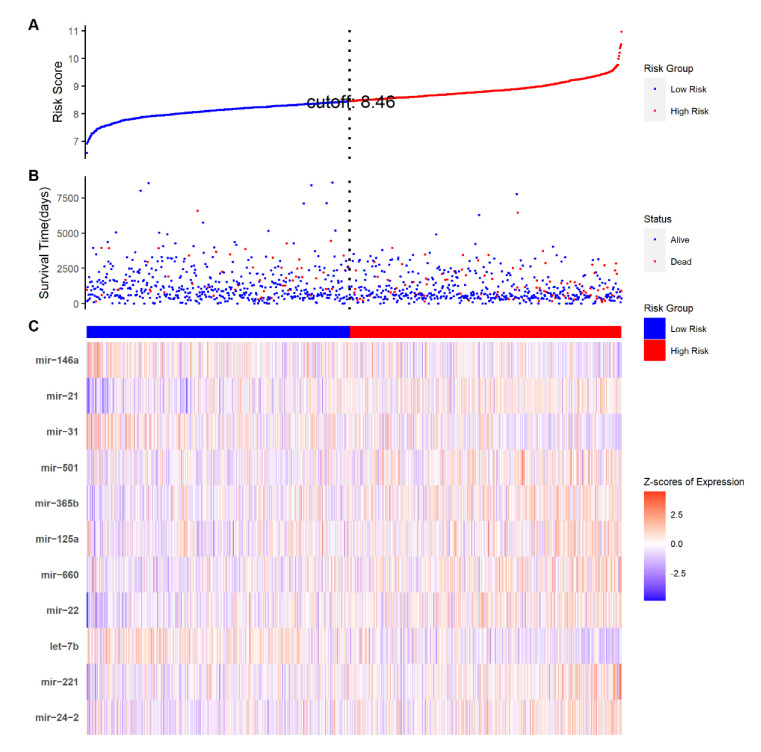
Risk score distribution and TRM expression heat map in TCGA-BRCA dataset. (**A**) Risk score distribution where blue dot signifies low-risk group and red dot signifies high-risk group. Vertical dotted lines indicate the cut-off point for median risk score. (**B**) Survival time and status for all patients. (**C**) Heat map of the eleven selected TRM expression in the TRM signature.

**Figure 4 ijms-23-06994-f004:**
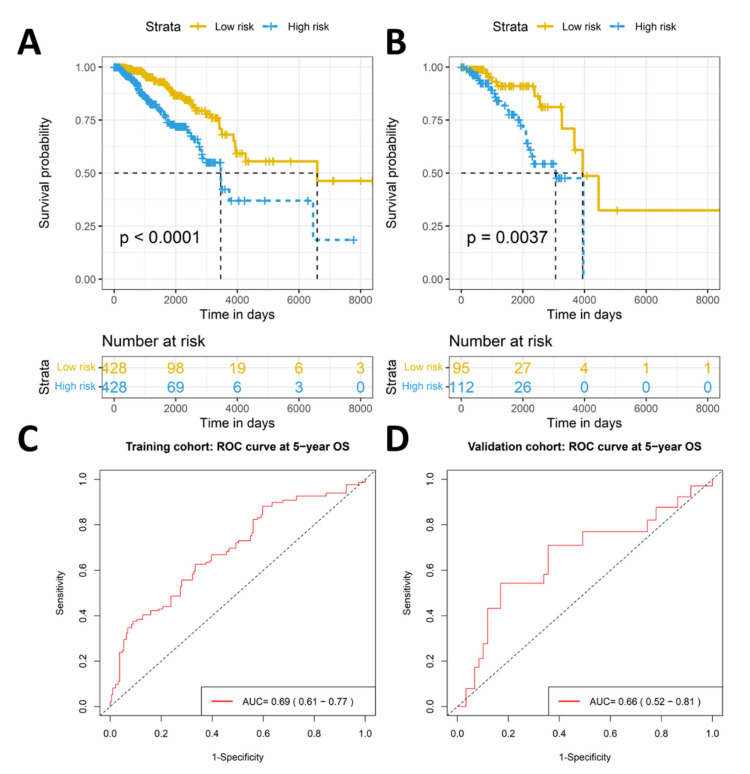
Kaplan–Meier survival analysis and time dependent ROC curves of the risk groups based on TRM signature for training and validation cohorts. KM curves show that the low-risk group has significantly longer overall survival compared to the high-risk group in both the training (**A**) and validation (**B**) cohorts. The AUCs of time-dependent ROC curves at 5-year OS were 0.69 and 0.66 for the training (**C**) and validation (**D**) cohorts, respectively.

**Figure 5 ijms-23-06994-f005:**
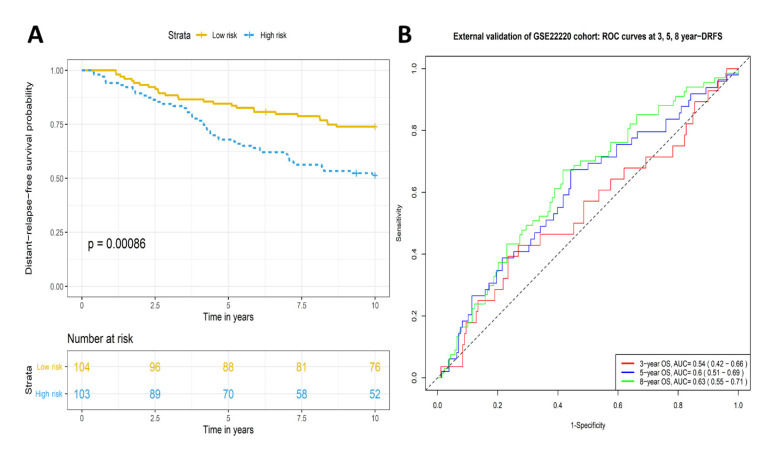
Kaplan–Meier survival analysis and time-dependent ROC curves of the risk groups based on the TRM signature for the external GEO GSE22220 cohort. (**A**) Patients of the high-risk group had significantly shorter DRFS (*p* value < 0.001). (**B**) AUCs of time -dependent ROC curves at 3-, 5- and 8-year DRFS were 0.54, 0.60 and 0.63, respectively.

**Figure 6 ijms-23-06994-f006:**
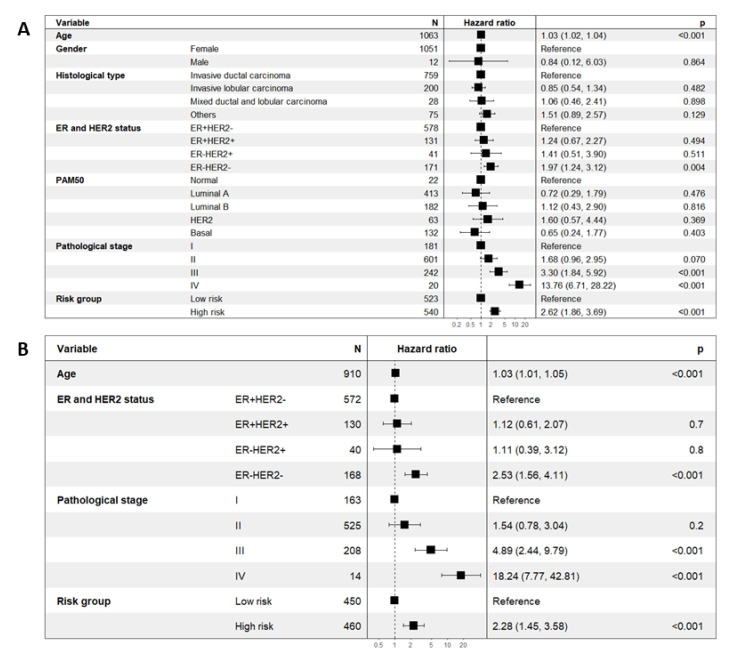
Univariate and multivariate Cox proportional hazards regression analysis for pertinent clinicopathological parameters. (**A**) Higher pathological stages (stage III and IV), ER and HER2 negative status, higher age and high-risk groups were significant poor prognostic factors in univariate analysis. (**B**) High-risk group remains an independent poor prognostic factor (*p* value < 0.001) in the multivariate analysis.

**Figure 7 ijms-23-06994-f007:**
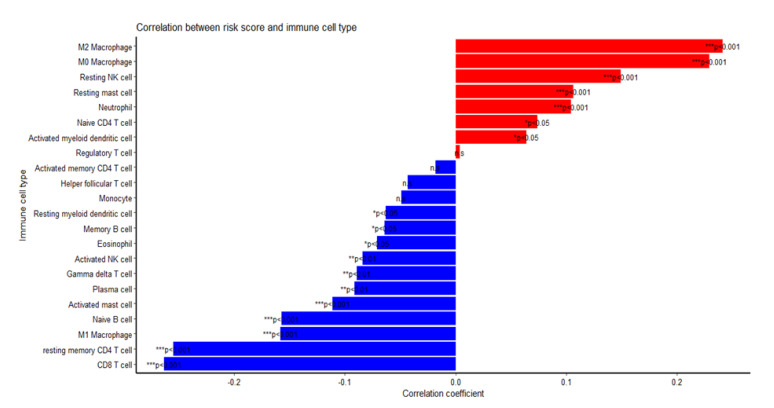
Correlation between risk score and immune infiltrate populations. M2 macrophages had the highest positive correlation with the risk score, while M1 had the third highest negative correlation, after CD8 T cells and CD4 T cells. * *p* < 0.05, ** *p* < 0.01, *** *p* < 0.001.

**Figure 8 ijms-23-06994-f008:**
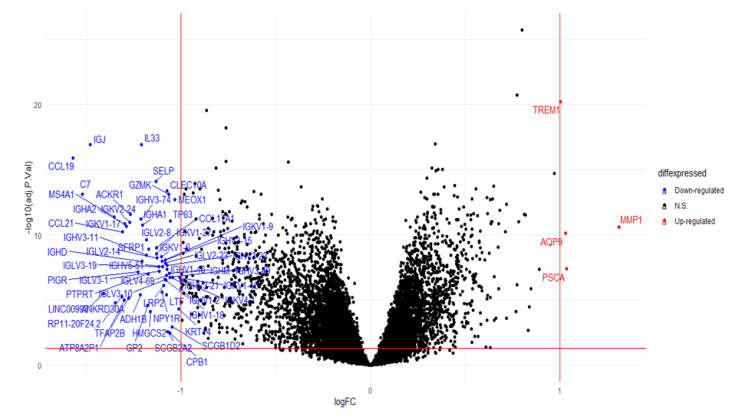
Volcano plot for the differential gene expression. Four genes were upregulated while 59 genes were downregulated in the high-risk group as compared to the low-risk group with the adjusted *p* value <  0.05 and |Log2(fold change)| > 1.

**Figure 9 ijms-23-06994-f009:**
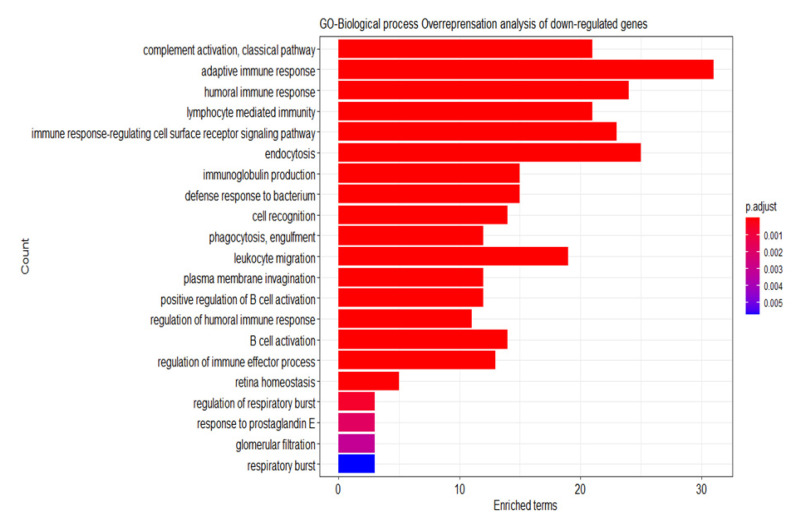
Overrepresentation analysis of the 59 downregulated genes by GO–biological process. The downregulated genes were mostly concentrated in the immune pathways, such as classical pathway of complement activation, adaptive immune response and humoral immune response.

**Figure 10 ijms-23-06994-f010:**
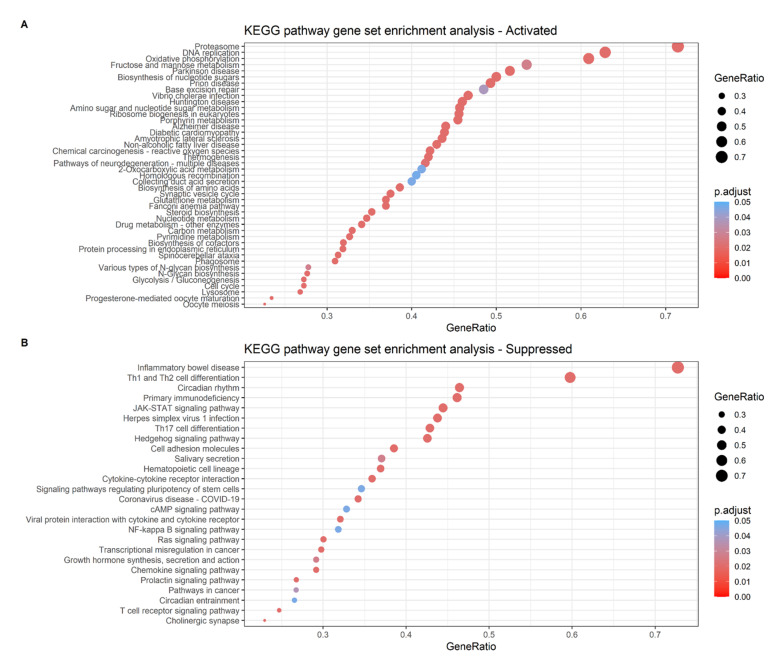
KEGG pathway gene set enrichment analysis for the ranked genes. Size of dots represents the GeneRatio while the color represents the *p* value. (**A**) Proteosome, DNA replication, oxidative phosphorylation and base excision repair pathways were significantly activated in the high-risk group as compared to the low-risk group. (**B**) Immunity pathways such as Th1 and Th2 cell differentiation, JAK-STAT signaling pathway and Th17 cell differentiation were suppressed in the high-risk group as compared to the low-risk group.

**Figure 11 ijms-23-06994-f011:**
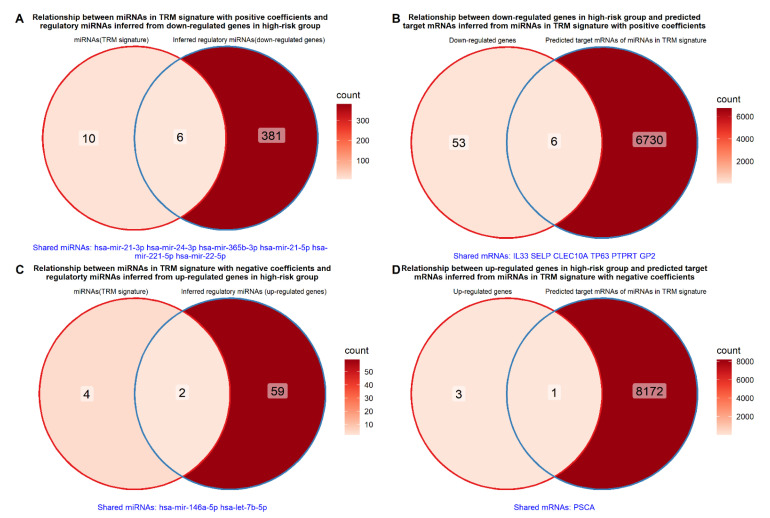
(**A**,**C**) Venn diagrams displaying relationship between miRNAs in the TRM signature and regulatory miRNAs inferred from the differentially expressed genes. (**B**,**D**) Venn diagrams displaying relationship between differently expressed genes between the high- and low-risk group and predicted target mRNAs of the miRNAs in the TRM signature.

**Table 1 ijms-23-06994-t001:** List of TAM-related miRNAs from various cancers.

No	miRNA	Cancer Type	Function	Ref	Precursor miRNAs
**1**	let-7a	Lung	transferred from TAMs to lung cancer to inhibit cell proliferation, migration, and invasion	[13]	let-7a-1 let-7a-2 let-7a-3
**2**	let-7b	Breast Prostate	repolarizes M2 TAMs to M1 in tumor cells modulates macrophage polarization to promote angiogenesis and mobility	[14] [15]	
**3**	miR-7	Ovarian	released by TAMs to inhibit cell metastasis	[16]	mir-7-1 mir-7-2 mir-7-3
**4**	miR-9	HNSCC	induces M1 TAM polarization and increases tumor radiosensitivity	[17]	mir-9-1 mir-9-2 mir-9-3
**5**	miR-15b	HCC	derived from M2 TAMs to promote cancer progression	[18]	
**6**	miR-16	Gastric	transferred from M1 TAMs to cancer cells to inhibit tumor formation	[19]	mir-16-1 mir-16-2
**7**	miR-18a	Nasopharynx Liver	derived from M2 TAMs to promote cancer progression and tumor growth induces M1 TAMs to inhibit tumor metastasis	[20] [21]	
**8**	miR-19a	Breast Gastric	downregulates M2 TAMs to inhibit cancer progression and metastasis derived from M2 TAMs to reduce chemosensitivity and tumor cell apoptosis	[22] [23]	
**9**	miR-21	Bladder Breast	promotes cancer progression by polarizing TAMs to M2 phenotype promotes M2 TAM transformation to induce metastasis	[24] [25]	
**10**	miR-22	Glioma	derived from TAMs to promote mesenchymal phenotype and induce radiotherapy resistance	[26]	
**11**	miR-23a	Breast	regulates TAM polarization	[27]	
**12**	miR-24-2	Breast	regulates M1 and M2 TAM polarization	[27]	mir-24-1 mir-24-2
**13**	miR-26a	Esophageal	M2 TAMs downregulate miR-26a to promote invasion and metastasis of cancer	[28]	mir-26a-1 mir-26a-2
**14**	miR-27a	Glioma	derived from TAMs to promote mesenchymal phenotype and induce radiotherapy resistance	[26]	
**15**	miR-29a	OSCC Ovarian	promotes M2 TAMs polarization to enhance proliferation and invasion of cancer cells derived by TAM to facilitate cancer cell proliferation and immune escape	[29] [30]	
**16**	miR-31	OSCC	derived by M2 TAMs to facilitate cancer progression	[31]	
**17**	miR-92a	Breast Liver	suppresses the infiltration of TAMs in tumor cells derived from TAMs to increase liver cancer cells invasion	[32] [33]	mir-92a-1 mir-92a-2
**18**	miR-95	Prostate	derived by M2 TAMs to promote cancer progression	[34]	
**19**	miR-122	Pancreatic	M2 TAMs increases miR-122-5p expression to inhibit PC progression	[35]	
**20**	miR-125a	HCC	inhibits TAMs mediated in cancer stem cells	[36]	
**21**	miR-125b	HCC	inhibits TAMs mediated in cancer stem cells	[36]	mir-125b-1 mir-125b-2
**22**	miR-130a	Lung	suppresses the polarization of M2 TAMs and enhances M1 polarization	[37]	
**23**	miR-130b	Gastric	transferred from M2 TAM to promote survival, migration, invasion, and angiogenesis	[38]	
**24**	miR-142	HCC Glioblastoma	transferred from TAM to cancer cells to inhibit proliferation, tumor growth and invasion inhibits glioma growth and induces apoptosis in M2 TAMs	[39] [40]	
**25**	miR-146a	Breast Endometrial HCC	promotes M2 TAM expression inhibits M2 TAM polarization promotes M2 polarization	[41] [42] [43]	
**26**	miR-146b	Ovarian Bladder	inhibits the migration of endothelial cells promotes M2 TAM infiltration	[44] [45]	
**27**	miR-155	Esophageal Lung Colon	derived from TAMs to suppress cancer proliferation, migration, invasion and vasculature formation secreted by M2 TAMs to promote metastasis derived from M2 TAMs to promote cell migration and invasion	[46] [47] [48]	
**28**	miR-221	Ovarian Osteosarcoma Glioma	released from M2 TAMs to promote cancer cell proliferation and progression derived from M2 TAMs to aggravate cancer growth and metastasis derived from TAMs to promote mesenchymal phenotype and induce radiotherapy resistance	[49] [50] [26]	
**29**	miR-222	Breast Ovarian	delivered to TAMs to induce M2 polarization regulates polarization of M2 TAMs	[51] [52]	
**30**	miR-223	Ovarian Breast Gastric	derived from TAM to enhance tumor malignancy and chemoresistance released by M2 TAMs to promote cancer cell invasion derived by M2 TAMs to promote drug resistance	[53] [54] [55]	
**31**	miR-326	HCC	derived by M1 TAMs to inhibit cancer cell proliferation, colony formation, migration and invasion	[56]	
**32**	miR-365	Pancreatic	secreted by M2 TAMs to induce drug resistance and promote cancer progression	[57] [58]	mir-365a mir-365b
**33**	miR-487a	Gastric	derived from M2 TAMs to promote cancer proliferation and tumorigenesis	[59]	
**34**	miR-501	Pancreatic Lung	derived by M2 TAMs to inhibit tumor suppressor TGFBR3 gene and facilitate cancer development derived by M2 TAMs to promote cancer progression	[60] [61]	
**35**	miR-503	Breast	derived from TAMs to suppress cancer progression	[62]	
**36**	miR-660	Ovarian	upregulated in TAMs that promote cancer progression	[63]	
**37**	miR-720	Breast	inhibits M2 TAM polarization	[64]	
**38**	miR-877	Breast	increases expression in the late 4T1 tumor TAMs	[41]	
**39**	miR-940	Ovarian	induces M2 TAMs polarization	[65]	
**40**	miR-4291	Breast	downregulated in TAMs that promote cancer progression	[66]	
**41**	miR-5100	Breast	inhibits invasion and migration of cancer	[66]	
**42**	miR-5196	Breast	downregulated in TAMs that promote cancer progression	[66]	

HNSCC: Head and neck squamous cell carcinoma, HCC: Hepatocellular carcinoma, OSCC: Oral squamous cell carcinoma.

## Data Availability

The TCGA-BRCA dataset used in this study can be obtained from the UCSC Xena exploration tool (https://xenabrowser.net/ (accessed on 28 August 2021)). GEO dataset used in this study can be obtained from the GEO database (https://www.ncbi.nlm.nih.gov/gds (accessed on 21 April 2022)).

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
