# Peer review of "An Eleven-microRNA Signature Related to Tumor-Associated Macrophages Predicts Prognosis of Breast Cancer"

_ijms, 2022, doi:10.3390/ijms23136994_

Round 1

Reviewer 1 Report

This manuscript aims to explore the prognostic value of TRMs in breast cancer, through the identification of a novel TRM signature of eleven miRNAs.

The study, although prospective, is interesting, could be a starting point for subsequent works, with experimental validations. In  addition, the methods used conform to the analysis carried out.

The manuscript can be accepted for publication if the authors are ready to incorporate the following revisions:

- The figure 1 appears unclear and not very visible, it should be graphically improved. I also recommend moving it to the 'results' section.

- The figures on the whole seem to be too many, some seem repetitive; I recommend selecting the most relevant and moving the others to supplementary materials.

- In paragraph 3., at line 223, 54 studies are indicated, while the table shows 42, please correct.

- For the completeness of the manuscript, it would be interesting to add to the discussion a part concerning the correlation between TAMs and methylation of specific genes breast cancer-related.

In this regard, they would benefit from reading the following articles, the contents of which could be useful for improving the manuscript:

1. PMID: 33092789 DOI: 10.1016/j.bbrc.2020.10.037

2. PMID: 34726251 DOI: 10.3892/ijo.2021.5278

Author Response

Reviewer 1

This manuscript aims to explore the prognostic value of TRMs in breast cancer, through the identification of a novel TRM signature of eleven miRNAs.

The study, although prospective, is interesting, could be a starting point for subsequent works, with experimental validations. In addition, the methods used conform to the analysis carried out.

The manuscript can be accepted for publication if the authors are ready to incorporate the following revisions:

Point 1: The figure 1 appears unclear and not very visible, it should be graphically improved. I also recommend moving it to the 'results' section.

Response 1: Thank you for the feedback. We have replaced and moved figure 1 to ‘Results’ section as suggested.

Point 2: The figures on the whole seem to be too many, some seem repetitive; I recommend selecting the most relevant and moving the others to supplementary materials.

Response 2: We thank you for the suggestion. We have moved figure 5, 8 and 13 (from previous manuscript) to supplementary material.

Point 3: In paragraph 3., at line 223, 54 studies are indicated, while the table shows 42, please correct.

Response 3: We thank you for the feedback. We would like to clarify that a total of 54 studies were indeed analyzed where 42 TRMs were eventually identified. This is because some studies reported the same TRM, but in different types of cancer.

Point 4: For the completeness of the manuscript, it would be interesting to add to the discussion a part concerning the correlation between TAMs and methylation of specific genes breast cancer-related.

In this regard, they would benefit from reading the following articles, the contents of which could be useful for improving the manuscript:

  1. PMID: 33092789 DOI: 10.1016/j.bbrc.2020.10.037
  2. PMID: 34726251 DOI: 10.3892/ijo.2021.5278

Response 4: Thank you very much for the kind suggestion and recommended articles. As suggested, we have now included a paragraph on breast cancer gene methylation and TAM in our discussion (Line 647-654).

Reviewer 2 Report

MiRNAs are increasingly recognized as important communication mediators in many diseases, including breast cancer. Similarly, macrophages are currently receiving considerable attention as important players in the tumor immune environment and in regulation of potential anti-tumor immunity. In this manuscript, the authors use in silico analyses to investigate relationships between miRNAs which have been previously implicated in macrophage activities (either secreted by cancer cells and acting on macrophages, or secreted by macrophages) and survival in breast cancer patients. They use literature-based approaches to identify miRNAs, noting that relatively little has been done with breast cancer and therefore they supplement with findings from other cancers. They then use the TCGA-BRCA database and one dataset from the GEO database to develop and test a TAM-related miRNA signature, and use various bioinformatic tools to predict immune infiltrates and processes affected. They conclude that they have developed a signature of 11 miRNAs that predicts overall and distant-relapse free survival in breast cancer.

This is an important area. The studies described extend our knowledge in this area and provide the basis for future investigations. However, this study would benefit by addressing several issues:

1.       There is a rich literature describing differences in the immune environment of the different breast cancer subtypes. The current ms has analyzed data from the different subtypes by both expression of ER and HER2, and also by molecular subtyping by PAM50.  

a.       A discussion of the differences in the immune TME across breast cancer subtypes would be useful.

b.       Is there a difference in predictive value of their signature among the subtypes?

c.       Regardless, discussion of their signature in the context of e.g., the greater lymphocytic infiltrate in TNBC tumors would be helpful.  

2.       They divide patients into “high risk” and “low risk” categories, splitting at the median. Would analysis of the highest and lowest quartiles provide a clearer picture?

3.       The authors make extensive use of the “M1/M2” macrophage polarization nomenclature to explain their findings. Indeed, the literature does focus on these states, although they were developed in the context of infectious disease, and a M1/M2 continuum is increasingly revealed by single cell sequencing in cancers, including breast cancer. It would be helpful to identify any miRNAs in their signature that may alter macrophage activity. Several statements in the ms may be addressing this issue, but would benefit from further clarification:  e.g., Lines 366-368: “…interrelationship between individual miRNAs and different immune infiltrate populations showed various strengths and directions of correlations without distinctive pattern with certain immune cell types”.  Also lines 573-577 in the discussion: “TRM signature and different immune infiltrate populations did not recapitulate such relationship at individual miRNA level…complex regulatory network”.  Doubtless this is true, but it is none-the-less puzzling that no direct regulatory pathways were identified.

Author Response

Kindly see the attachment for our response.
